# Silica Production across Candidate Lignocellulosic Biorefinery Feedstocks

**Yifeng Xu [1], Nick Porter [1], Jamie L. Foster [2], James P. Muir [3], Paul Schwab [1], Byron L. Burson [4] and Russell W. Jessup [1,*]**

1   Department of Soil & Crop Sciences, Texas A&M University, College Station, TX 77843, USA; xuyfjhjy@tamu.edu (Y.X.); nick.porter1992@tamu.edu (N.P.); pschwab@tamu.edu (P.S.)
2   Texas A&M AgriLife Research Station, Beeville, TX 78102, USA; JLFoster@ag.tamu.edu
3   Texas A&M AgriLife Research Station, Stephenville, TX 76401, USA; Jim.Muir@ag.tamu.edu
4   USDA–ARS, Crop Germplasm Research Unit, Southern Plains Agricultural Research Center, College Station, TX 77843, USA; Byron.Burson@ARS.USDA.GOV
*   Correspondence: rjessup@tamu.edu; Tel.: +1-979-315-4242

**Abstract:** Biofuels produced from non-food lignocellulosic feedstocks have the potential to replace a significant percentage of fossil fuels via high yield potential and suitability for cultivation on marginal lands. Commercialization of dedicated lignocellulosic crops into single biofuels, however, is hampered by conversion technology costs and decreasing oil prices. Integrated biorefinery approaches, where value-added chemicals are produced in conjunction with biofuels, offer significant potential towards overcoming this economic disadvantage. In this study, candidate lignocellulosic feedstocks were evaluated for their potential biomass and silica yields. Feedstock entries included pearl millet-napiergrass ("PMN"; *Pennisetum glaucum* [L.] R. Br. × *P. purpureum* Schumach.), napiergrass (*P. purpureum* Schumach.), annual sorghum (*Sorghum bicolor* [L.] Moench), pearl millet (*P. glaucum* [L.] R. Br.), perennial sorghum (*Sorghum* spp.), switchgrass (*Panicum virgatum* L.), sunn hemp (*Crotalaria juncea* L.), giant miscanthus (*Miscanthus × giganteus* J.M. Greef and Deuter), and energy cane (*Saccharum* spp.). Replicated plots were planted at three locations and characterized for biomass yield, chemical composition including hemicellulose, cellulose, acid detergent lignin (ADL), neutral detergent fiber (NDF), crude protein (CP), and silica concentration. The PMN, napiergrass, energy cane, and sunn hemp had the highest biomass yields. They were superior candidates for ethanol production due to high cellulose and hemicellulose content. They also had high silica yield except for sunn hemp. Silica yield among feedstock entries ranged from 41 to 3249 kg ha$^{-1}$. Based on high bioethanol and biosilica yield potential, PMN, napiergrass, and energy cane are the most promising biorefinery feedstock candidates for improving biofuel profitability.

**Keywords:** biofuels; biorefinery; silica; biosilica

---

## 1. Introduction

### 1.1. Integrated Biorefineries

Production of biofuels as a single revenue source remains economically unprofitable [1]. The Environmental Protection Agency's (EPA) Renewable Fuel Standard (RFS) program, in consultation with U.S. Department of Agriculture and the Department of Energy, mandated a long-term goal to produce 136 billion liters of renewable fuel by 2022. Approximately 61 billion liters of this total was to be produced from cellulosic biofuels [2]. However, the EPA significantly reduced the volume requirement for cellulosic biofuel in 2017 from 21 to only 1.2 billion liters [3].

Biorefinery approaches that diversify output streams by generating both primary biofuels and value-added co-products have potential for increased profitability. Conceptual extensions of lignocellulosic feedstocks from current single biofuel platforms to integrated biorefineries involve separation and utilization of compositional fractions of biomass into primary biofuels (ethanol from cellulose and hemicellulose, for example) and additional bioproducts from the remaining lignin (lignosulfonates, bioplastics, etc.) [4] and mineral (biosilica, etc.) fractions.

Depolymerization of plant biomass results in primary fractions of cellulose, hemicellulose, and lignin. Both cellulose and hemicellulose are polysaccharides, but they differ in their primary components and structures. Cellulose is composed of a linear and unbranched chain of β-(1,4)-linked D-glucose, while hemicellulose can be classified as xylan, xyloglucan, glucuronoxylan, arabinoxylan, or glucomannan based on its branched chain [5,6]. Cellulases, the enzymes necessary to break β-(1,4) bonds, hydrolyze cellulose to glucose. However, a complex enzymatic cocktail is needed to hydrolyze hemicellulose to pentose [6]. Once the monosaccharide is obtained, downstream fermentation produces ethanol. Unlike cellulose and hemicellulose, lignin is a polyphenolic polymer and is often treated as a hindrance for efficient biomass conversion [6]. Most of the lignin is directly combusted for the production of energy during the pulping process, and only small amounts have been utilized for conversion into other chemicals. If different pulp processing methods are used, lignosulfonates can be produced in leu of lignin being directly combusted. The largest volumes of lignosulfonates (50–90%) are utilized as active plasticizing agents in concrete admixture systems as a cost-efficient alternative to synthetic superplasticizers that are derived from fossil fuels [7]. To date, the utilization of lignin either directly for biopower or indirectly by producing lignosulfonates has not proven sufficient towards making biofuel refineries profitable. Investigation of additional, value-added co-products that can be obtained from the residual mineral fraction is therefore warranted.

## 1.2. Silica

One alternative co-product with potential to increase the profitability of biofuel refining is amorphous silica ($SiO_2$). This material is used in diverse industrial products such as: semiconductors, nanotechnology, reinforcing agents, as a filler, and in specialty chemicals. The majority of silica is produced today through the smelting of quartz at high temperature; however, a relatively energy efficient method of isolating silica has been demonstrated using rice hull ash [8]. This methodology could be incorporated into current dedicated biofuel conversion strategies, in which the residual mineral fraction is not utilized.

Silica within a plant depends on its uptake from the soil in the form of soluble $Si(OH)_4$ or $Si(OH)_3O^-$. It is ubiquitous across plants, ranging from 0.1 to more than 10% dry weight [9]. Grasses contain among the highest silica concentrations, which varies within different parts of a plant. In rice (*Oryza sativa* L.), for example, silica content reaches 13% in straw, 23% in hulls, and 35% in nodes [10]. Silica concentrations in perennial grasses such as guineagrass (*Panicum maximum* Jacq.) (1.07%) and napiergrass (0.85%) are higher than in sugarcane (*Saccharum officinarum* L.) bagasse (0.44%) [11]. Silica concentration in napiergrass can vary between 0.57% and 4.21%, and higher values are found in the leaves rather than in the stems [12]. Drought stress further induces silica accumulation, resulting in silica concentrations in napiergrass blades and sheaths up to 5% and 3.4%, respectively [13]. High silica-concentration napiergrass ash has been investigated for its use in many applications, including glass manufacturing and clay ceramics [14,15]. The reported median value of silica concentrations in switchgrass samples (1.5%) is higher than that for *M. × giganteus* (1.08%) [16]. A two-step process to isolate lignin and silica from biomass derived black liquor—a waste product produced in the pulping process that is high in lignin and other extracts—revealed that high silica recovery in the precipitate could be achieved at pH 6–7. Below this pH range, silica was re-dissolved into the solution [17].

### 1.3. Feedstocks

A large and diverse collection of high-biomass feedstocks was utilized, including six perennial grasses (napiergrass, pearl millet-napiergrass, switchgrass, energycane, miscanthus, and perennial sorghum), two annual grasses (pearl millet and annual sorghum), and one legume (sunn hemp). Napiergrass is a robust perennial forage grass that produces more biomass than most other grasses [18,19], ranging from 8.3 to 27.3 Mg ha$^{-1}$ y$^{-1}$ [20–22].

Napiergrass (2n = 4x = 28) will hybridize with pearl millet (2n = 2x = 14) to produce interspecific triploid hybrids (2n = 3x = 21). These hybrids combine the forage quality of pearl millet [18], comparatively large seed size and seed yield of annual grasses such as sorghum (*Sorghum bicolor* [L.] Moench) [20–24], lower establishment costs than energycane and giant miscanthus (*Miscanthus* × *giganteus*) [25], biomass yields as high as 37 Mg DM ha$^{-1}$ y$^{-1}$ in subtropical climates [26].

Pearl millet (*Pennisetum glaucum* [L.] R. Br.) is an annual diploid (2n = 2x = 14) grass of African origin [27] utilized worldwide as a grain crop, a forage crop [28], or a high biomass feedstock [29,30].

Switchgrass (*Panicum virgatum* L.), a perennial grass indigenous to North America, can be utilized as either a forage bioenergy crop [31,32] with wide range of adaptation, genetic diversity, and suitability for marginal lands. The cultivar 'Alamo' has higher biomass yield [33], and with N application of at least 150 kg N ha$^{-1}$ y$^{-1}$ can achieve 10 to 15 Mg DM ha$^{-1}$ y$^{-1}$ [34–36].

Giant miscanthus (*Miscanthus* × *giganteus)* is a sterile, triploid (2n = 3x = 57), perennial interspecific hybrid between *M. sinensis* Andersson (2n = 2x = 38) and *M. sacchariflorus* (Maxim.) Hackel (2n = 2x = 76) [37]. Biomass yields, excluding the two establishment years, range from 22.0 to 35.4 Mg ha$^{-1}$ y$^{-1}$ in temperate environments, are significantly lower in subtropical regions [38,39], and are generally higher than switchgrass [40,41].

Energy cane (*Saccharum* L. spp.) is a perennial bioenergy crop derived from sugarcane, possessing higher fiber concentration, higher biomass yields, and better cold tolerance [42–44]. It is productive on marginal lands [45] and has biomass yields comparable to other lignocellulosic feedstocks [46].

Sorghum is used mainly for grain and forage production, but recently it has been evaluated as a bioenergy crop [47]. Chemically induced high-value brown-midrib mutants in sorghum improve forage quality [48], reduce lignin concentration as much as 51% in stems and 25% in leaves [49], and improve overall cellulosic ethanol conversion efficiency [50].

Sunn hemp (*Crotalaria juncea* L.) is a legume native to India used for soil restoration, green manure, and livestock feed [51,52]. High biomass yields and significant N contributions to subsequent crops make sunn hemp an alternative cover crop in warm temperate regions [53] with potential to replace winter legumes as cover crops [54,55]. It has been demonstrated to produce 10.7 Mg DM ha$^{-1}$ after 12 weeks of growth which is equivalent to 204 GJ ha$^{-1}$ energy yield [56].

### 1.4. Rationale

Direct, side-by-side comparative biomass yield evaluations of lignocellulosic feedstocks are limited. Those available are further lacking in both data across multiple adaptation regions as well as biofuel: co-product yields. As a result, extrapolation of feedstock performance towards specific biofuel conversion strategies that also include value-added bioproducts from inorganic, mineral fraction remains difficult. The objective of this study was therefore to evaluate nine diverse biomass crops across multiple ecoregions for biomass yield, forage composition, and both silica content and yield. This would provide the first report for Si content and yield among diverse candidate biofuel feedstocks across temperate ecoregions.

## 2. Materials and Methods

### 2.1. Plant Entries

Twelve feedstocks, including seven grass species and one legume species, were evaluated (Table 1).

**Table 1.** Feedstocks utilized for field trials.

| Entry | Species | Identification | Life Cycle | Family |
|---|---|---|---|---|
| 1 | Pearl Millet-Napiergrass (PMN) | PMN10TX13 | Perennial | Poaceae |
| 2 | Napiergrass | Merkeron | Perennial | Poaceae |
| 3 | Napiergrass | PEPU 09FL03 | Perennial | Poaceae |
| 4 | Napiergrass | PEPU 09FL01 | Perennial | Poaceae |
| 5 | BMR sorghum | SDH2942 | Annual | Poaceae |
| 6 | Annual sorghum | SX-17 | Annual | Poaceae |
| 7 | BMR Pearl millet | Exceed | Annual | Poaceae |
| 8 | Perennial sorghum | PSH 09TX15 | Perennial | Poaceae |
| 9 | Switchgrass | Alamo | Perennial | Poaceae |
| 10 | Sunn hemp | Tropical Isle | Annual | Fabaceae |
| 11 | Giant miscanthus | (Mxg) | Perennial | Poaceae |
| 12 | Energy cane | (unknown accession) | Perennial | Poaceae |

## 2.2. Field Evaluation

### 2.2.1. Propagation of Plant Materials and Planting

Culms of the napiergrass (PEPU 09FL01, PEPU 09FL03, Merkeron) and energy cane entries were harvested in October 2015 from field plots at College Station, TX. Single nodes were cut from the stalks and planted into a soil mixture in 95 L barrels inside a greenhouse to grow for propagation into trays in spring 2016. Rhizomes of PSH 09TX15 perennial sorghum and giant miscanthus were collected at the same time and similarly increased. In April 2016, individual plants of napiergrass, energy cane, perennial sorghum, and giant miscanthus lines were removed from the barrels and transplanted into a commercial soil mix in propagation trays and allowed to acclimate outside for 4 wk. Single switchgrass and PMN seed were seeded directly into propagation trays, established for 8 wk, and allowed to acclimate outside for 4 wk.

In May 2016, replicated plots (n = 3) were planted in a completely randomized design at College Station, Beeville, and Stephenville, TX. The College Station location (30°32′ N, 96°26′ W; elevation 81m) was on a Weswood silty clay loam (pH 8.0). The Beeville location (28°27′ N, 97°42′ W; elevation 70 m) was on a Parrita sandy clay loam (pH 7.2). The Stephenville location (34°17′ N, 96°12′ W; elevation 370 m) was on a Windthorst fine sandy loam (pH 6.8). Each cultivar was planted in three plots (3 × 3 m) with four, 3-m rows at 0.75 m between plants within a row. Propagated entries (1, 2, 3, 4, 8, 9, 11, and 12) were planted vegetatively with seven plants in each row. At Beeville and College Station, entries 1 and 9 had been planted for a previous experiment in 2014. Entries 5, 6, 7, and 10 were planted by seed at 2.5 cm spacing within rows using a Jang JP-1 roller-type seeder. For the 2017 growing season, vegetatively propagated entries were regrown from 2016 field plots in each three location. Entries 5, 6, 7, and 10 were again planted in 2016 using a Jang JP-1 roller-type seeder. Weed control was conducted by hand and mechanical cultivation. A single fertilizer application of 80 kg N per hectare (urea) was applied 3 wk post planting on all grass plots in both growing seasons. Total water inputs (precipitation and irrigation) varied across trial sites and were adequate for high biomass yield potential (Figure 1).

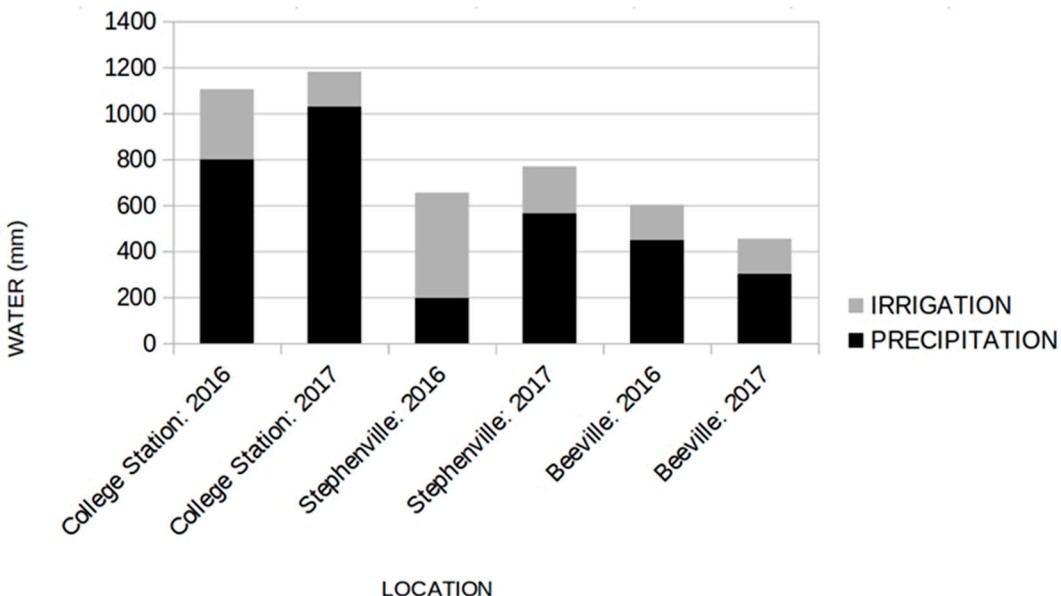

**Figure 1.** Total cumulative irrigation and precipitation inputs (mm) at trial sites in College Station, Stephenville, and Beeville, TX in 2016 and 2017.

### 2.2.2. Harvesting and Estimation of Biomass Yield

Field harvests were made in November 2016 and 2017. One of the two center rows of each plot was harvested, and the wet weight was measured. A subsample was obtained from each plot, weighed, air dried, and reweighed to determine moisture content and biomass yield. The air-dried subsamples were first ground with a hammer mill and then ground with a Wiley mill (Thomas Scientific, Philadelphia, PA, USA) through a 1-mm sieve for chemical composition and bio-silica analyses.

### 2.2.3. Chemical Composition (NDF, ADF, ADL, CP, Cellulose, Hemicellulose)

Neutral detergent fiber (NDF), acid detergent fiber (ADF), and acid detergent lignin (ADL) were determined using the methodology described by Van Soest and Robertson [57], modified by using an Ankom 200 Fiber Analyzer (Ankom Technologies, Macedon, NY, USA). Acid detergent lignin is the residue remaining after sequential digestion of ADF residue with 72% sulfuric acid [58]. Nitrogen (N) was determined using an elemental analyzer (Vario Macro, Elementar, Germany) and crude protein (CP), cellulose, and hemicellulose were calculated (CP = N, % DM basis × 6.25) (Cellulose = ADF − ADL) (Hemicellulose = NDF − ADF).

### 2.2.4. Silica Analysis

Silica was measured according to Reidinger et al. [59] using a portable X-ray fluorescence spectrometer (DELTA Premium, OLYMPUS, Tokyo, Japan). This method requires relatively small amounts of plant material and is an accurate and rapid technique for detecting silica content in plant tissue [59]. To test the accuracy of this methodology, silica calibration standards were made by first mixing methyl cellulose and silica powder and then homogenizing them to produce standards with 0%, 0.5%, 1%, 2%, 3%, 4%, 5%, 6%, 7%, 8%, 9%, and 10% silica concentration. Pellets of both the silica calibration standards and the dried, ground plant samples were made by pressing 1.0 g of each sample into a 2 cm die with 12 Mg pressure using a manual hydraulic press. The pellets were subsequently analyzed for silica content using the X-ray fluorescence spectrometer.

### 2.3. Data Analysis and Statistics

The experimental units for data analysis were individual plots. The statistical model consisted of location, year, and plant entry in a three-factorial arrangement looking at three-way interactions and,

if those were not significant, at simple effects. Data were submitted to analysis of variance and when significant separated using All Pair, Tukey HSD with JMP software (JMP Pro12.1, Statistical Analysis System, Raleigh, NC, USA). Differences were considered significant at $p \leq 0.05$.

## 3. Results

### 3.1. Summary Statistics

There were differences among locations for all traits evaluated (Table 2). Year and feedstock entry effects varied. Interactions between location and year occurred for every trait. Location by feedstock entries interacted across all traits except for ADL. Year by feedstock entries interacted across all traits except for hemicellulose. Three-way interaction between location, year, and feedstock entries were nonsignificant.

**Table 2.** Analysis of variance of biomass yield, hemicellulose, cellulose, acid detergent lignin (ADL), neutral detergent fiber (NDF), crude protein (CP), and silica concentration.

| | Biomass | NDF | Hemicellulose | Cellulose | ADL | CP | Silica |
|---|---|---|---|---|---|---|---|
| location (loc) | * Z | *** | *** | *** | *** | *** | *** |
| year | *** | *** | * | *** | ns | *** | *** |
| treatment (trt) | *** | *** | *** | *** | *** | *** | *** |
| loc * year | *** | *** | *** | *** | *** | *** | *** |
| loc * trt | *** | *** | * | *** | ns | * | * |
| year * trt | *** | *** | ns | *** | * | *** | * |
| loc * year * trt | ns | ns | ns | ns | ns | ns | ns |

Z ns (nonsignificant) or significant at $p \leq 0.05$ (*), 0.01 (**), or 0.001 (***).

### 3.2. Year 1: Biomass Yield and Chemical Composition

At Stephenville, energy cane had the highest yield but was closely followed by PMN, all three napiergrass entries, both annual sorghum entries, and sunn hemp (Table 3). Giant miscanthus and pearl millet had the lowest yields. Despite being in its third season from a previous experiment, switchgrass yields were lower. At College Station, third season PMN had the highest yield (Table 4). Two napiergrass entries (Merkeron, PEPU 09FL03), energy cane, and sunn hemp had intermediate yields at this location (Table 4). All remaining entries, including third season switchgrass, had lower yields. The third season PMN entry at Beeville had the highest yield for the first harvest (Table 5), and part of the reason for this high yield was these plants were established two years earlier for another experiment. There were no significant differences among the remaining entries at Beeville (Table 5). Of the 11 remaining entries, all three napiergrass entries, the giant miscanthus entry, the energy cane entry, and one sorghum accession were the most productive (Table 5), The sunn hemp entry was heavily defoliated by rabbits, and insufficient forage was available to determine forage production yield.

Giant miscanthus had the highest hemicellulose concentration at College Station and Beeville (Tables 4 and 5), while at Stephenville pearl millet had the highest hemicellulose concentration (Table 3). At all locations, sunn hemp had the lowest hemicellulose concentration. Sunn hemp had the highest cellulose concentration at Stephenville and College Station. Across most remaining entries, cellulose concentration was highly uniform at all locations. Sunn hemp had the highest ADL at all locations. Two napiergrass entries (PEPU 09FL01, PEPU 09FL03) and PMN were statistically equivalent to sunn hemp for ADL at Beeville. The BMR sorghum had the lowest ADL at College Station and Beeville, while BMR pearl millet had the lowest ADL at Stephenville. The perennial sorghum entry had the highest NDF at Stephenville and Beeville, while the NDF of the non-BMR sorghum was the highest at College Station. The NDF of pearl millet was the lowest at Stephenville and College Station; however, BMR sorghum had the lowest NDF at Beeville. Crude protein in sunn hemp was higher than the

other feedstock entries across all locations. Across all locations, BMR sorghum had the highest silica concentration, while sunn hemp was the lowest in silica concentration. However, the silica yield (kg ha$^{-1}$) of the BMR sorghum tended to range from intermediate to low at all locations, and the silica yield of perennial sorghum and switchgrass was equivalent to the BMR sorghum. The silica yield of the third year PMN was the highest at College Station and Beeville. At Stephenville, even during its first year, the PMN possessed the highest silica yield but similar yield to one napiergrass accession (PEPU 09FL03) and the energy cane.

**Table 3.** First year Stephenville sample traits of biomass yield (kg ha$^{-1}$) and chemical composition including hemicellulose, cellulose, acid detergent lignin (ADL), neutral detergent fiber (NDF) (g kg$^{-1}$ biomass), crude protein (CP) (%), silica (%), and silica yield (kg ha$^{-1}$).

| | | | | Means | | | | |
|---|---|---|---|---|---|---|---|---|
| Entry | Biomass | Hemicellulose | Cellulose | ADL | NDF | CP | Silica | Silica Yield |
| | kg ha$^{-1}$ | | g kg$^{-1}$ | | | % | | kg ha$^{-1}$ |
| (1) PMN 10TX13 [Y] | 30,975 ab [Z] | 247 c | 404 bc | 64.2 abc | 716 abcd | 4.06 bcd | 4.11 ab | 1281 a |
| (2) Merkeron [X] | 23,589 abc | 245 c | 398 bc | 68.9 abc | 712 abcd | 3.44 bcd | 2.68 c | 631 abc |
| (3) PEPU 09FL03 [W] | 32,469 ab | 261 bc | 378 bcde | 71.5 abc | 711 bcd | 3.06 cd | 3.58 bc | 1264 ab |
| (4) PEPU 09FL01 [W] | 17,955 abc | 245 c | 402 bc | 87.4 ab | 734 abc | 2.69 d | 3.48 bc | 627 abc |
| (5) BMR Sorghum [V] | 8746 abc | 278 abc | 314 e | 70.3 abc | 662 cd | 4.44 abcd | 5.36 a | 468 abc |
| (6) SX-17 [U] | 16,912 abc | 260 abc | 387 bcd | 67.4 abc | 715 abcd | 2.63 d | 3.54 bc | 577 abc |
| (7) BMR pearl millet [T] | 7804 c | 306 a | 333 de | 29.4 c | 668 d | 5 ab | 2.73 c | 212 c |
| (8) PSH 09TX15 [S] | 10,872 bc | 289 ab | 413 b | 65.1 abc | 767 a | 2.5 d | 4.91 a | 542 abc |
| (9) Alamo switchgrass | 11,717 bc | 297 ab | 369 bcde | 64.7 abc | 730 abc | 3.31 bcd | 2.98 c | 350 bc |
| (10) Tropical Isle Sunn Hemp | 31,774 ab | 144 d | 507 a | 98.5 a | 749 ab | 6.56 a | 0.4 d | 144 bc |
| (11) Giant miscanthus (Mxg) | 4453 c | 285 abc | 356 cde | 64.9 abc | 705 bcd | 4.75 abc | 4.42 ab | 198 c |
| (12) Energy cane [R] | 33,044 a | 260 bc | 371 bcde | 50.1 bc | 681 cd | 3.38 bcd | 3.04 c | 1008 abc |

[Z] Means within a column under each main factor followed by the same letter (a,b,c,d,e) are not significantly different according to All Pairs Grouping, Tukey HSD. [Y] Pearl millet napiergrass hybrid PMN 10TX13. [X] Napiergrass accession. [W] Annual sorghum (BMR) cultivar. [V] Annual sorghum SX-17 cultivar. [U] Perennial sorghum hybrid PSH 09TX15. [T] Exceed BMR pearl millet. [S] Perennial sorghum hybrid PSH 09TX15. [R] Energy cane unknown accession.

**Table 4.** First year College Station sample traits of biomass yield (kg ha$^{-1}$) and chemical composition including hemicellulose, cellulose, acid detergent lignin (ADL), neutral detergent fiber (NDF) (g kg$^{-1}$ biomass), crude protein (CP) (%), silica (%), and silica yield (kg ha$^{-1}$).

| | | | | Means | | | | |
|---|---|---|---|---|---|---|---|---|
| Entry | Biomass | Hemicellulose | Cellulose | ADL | NDF | CP | Silica | Silica Yield |
| | kg ha$^{-1}$ | | g kg$^{-1}$ | | | % | | kg ha$^{-1}$ |
| (1) PMN 10TX13 [Y] | 71,318 a [Z] | 216 b | 401 bcd | 97.2 ab | 714 ab | 2.25 d | 4.2 bcd | 3249 a |
| (2) Merkeron [X] | 16,442 bc | 241 ab | 417 bc | 67.5 cdef | 726 ab | 3.31 bcd | 3.85 bcd | 645 b |
| (3) PEPU 09FL03 [W] | 9461 bc | 235 ab | 374 cd | 90.7 abc | 700 ab | 2.6 bcd | 3.62 bcd | 311 b |
| (4) PEPU 09FL01 [W] | 6000 c | 258 ab | 375 cd | 69.6 cdef | 702 ab | 3.15 bcd | 4.22 bcd | 247 b |
| (5) BMR Sorghum [V] | 3338 c | 255 ab | 416 bc | 35.5 g | 706 ab | 3.25 bcd | 6.74 a | 238 b |
| (6) SX-17 [U] | 4316 c | 241 ab | 438 b | 70.4 cdef | 749 a | 2.41 cd | 3.98 bcd | 175 b |
| (7) BMR pearl millet [T] | 1567 c | 276 a | 358 d | 46.4 efg | 680 b | 4.51 abc | 3.53 cd | 53 b |
| (8) PSH 09TX15 [S] | 3961 c | 243 ab | 428 b | 71.5 cde | 742 ab | 3.07 bcd | 5.21 ab | 200 b |
| (9) Alamo switchgrass | 6015 c | 289 a | 372 cd | 77.6 bcd | 738 ab | 2.64 bcd | 2.93 d | 178 b |
| (10) Tropical Isle Sunn Hemp | 25,820 b | 120 c | 505 a | 105 a | 730 ab | 5.48 a | 0.49 e | 90 b |
| (11) Giant miscanthus (Mxg) | 2328 c | 289 a | 361 d | 56.8 defg | 707 ab | 4.69 ab | 4.61 bc | 106 b |
| (12) Energy cane [R] | 13,704 bc | 266 ab | 376 cd | 45 fg | 687 ab | 3.21 bcd | 3.77 bcd | 518 b |

[Z] Means within a column under each main factor followed by the same letter (a,b,c,d,e,f,g) are not significantly different according to All Pairs Grouping, Tukey HSD. [Y] Pearl millet napiergrass hybrid PMN 10TX13. [X] Napiergrass accession. [W] Annual sorghum (BMR) cultivar. [V] Annual sorghum SX-17 cultivar. [U] Perennial sorghum hybrid PSH 09TX15. [T] Exceed BMR pearl millet. [S] Perennial sorghum hybrid PSH 09TX15. [R] Energy cane unknown accession.

**Table 5.** First year Beeville sample traits of biomass yield (kg ha$^{-1}$) and chemical composition including hemicellulose, cellulose, acid detergent lignin (ADL), neutral detergent fiber (NDF) (g kg$^{-1}$ biomass), crude protein (CP) (%), silica (%), and silica yield (kg ha$^{-1}$).

| | | | | | Means | | | |
|---|---|---|---|---|---|---|---|---|
| **Entry** | **Biomass** | **Hemicellulose** | **Cellulose** | **ADL** | **NDF** | **CP** | **Silica** | **Silica Yield** |
| | **kg ha$^{-1}$** | | **g kg$^{-1}$** | | | **%** | | **kg ha$^{-1}$** |
| (1) PMN 10TX13 [Y] | 69,519 Za [Z] | 232 d | 422 a | 74.9 abc | 728 ab | 5.86 b | 3.11 de | 2211 a |
| (2) Merkeron [X] | 18,955 b | 255 abcd | 414 ab | 65.2 bcd | 734 ab | 3.82 bcd | 4.38 abcd | 825 b |
| (3) PEPU09FL03 [W] | 15,635 b | 250 abcd | 388 abcd | 90.9 ab | 729 ab | 3.35 cd | 4.15 bcd | 639 b |
| (4) PEPU09FL01 [W] | 12,803 b | 244 bcd | 414 ab | 99.6 a | 758 a | 3.1 cd | 3.86 cd | 494 b |
| (5) BMR Sorghum [V] | 6678 b | 240 cd | 367 bcd | 40.5 d | 648 c | 3.92 bcd | 6.05 a | 412 b |
| (6) SX-17 [U] | 13,260 b | 239 cd | 400 abc | 62.7 bcd | 701 b | 3.32 cd | 3.65 de | 473 b |
| (7) BMR pearl millet [T] | 919 b | 278 abc | 380 abcd | 46.5 cd | 705 abc | 4.94 bcd | 4.4 abcd | 41 b |
| (8) PSH 09TX15 [S] | 9961 b | 265 abcd | 433 a | 61.8 bcd | 760 a | 2.6 d | 5.58 abc | 559 b |
| (9) Alamo switchgrass | 6137 b | 282 ab | 359 cd | 62.5 bcd | 704 b | 3.53 bcd | 2.97 de | 181 b |
| (10) Tropical Isle Sunn Hemp | - | 115 e | 439 ab | 111 a | 665 bc | 10.3 a | 1.03 e | - |
| (11) Giant miscanthus (Mxg) | 1624 b | 289 a | 348 d | 52.9 cd | 690 bc | 5.34 bc | 5.83 ab | 100 b |
| (12) Energy cane [R] | 14,239 b | 266 abcd | 383 abcd | 57.2 cd | 706 b | 4.26 bcd | 4.3 abcd | 613 b |

[Z] Means within a column under each main factor followed by the same letter (a,b,c,d) are not significantly different according to All Pairs Grouping, Tukey HSD. [Y] Pearl millet napiergrass hybrid PMN 10TX13. [X] Napiergrass accession. [W] Annual sorghum (BMR) cultivar. [V] Annual sorghum SX-17 cultivar. [U] Perennial sorghum hybrid PSH 09TX15. [T] Exceed BMR pearl millet. [S] Perennial sorghum hybrid PSH 09TX15. [R] Energy cane unknown accession.

### 3.3. Year 2: Biomass Yield and Chemical Composition

The second-year biomass production was greater than that of the first-year yields for all perennial feedstocks except for PMN at all locations and PSH 09TX15 at Beeville (Tables 6–8). At Stephenville, the PMN dry matter yield decreased the most from year one to year two. This occurred because the PMN did not overwinter well in Stephenville and there were fewer PMN plants in the plot the second year. Even with reduced yields, PMN 10TX13 had the highest biomass yield at Beeville and College Station, but at both locations no significant differences were found for biomass yield among the PMN, energy cane, or napiergrass entries. Except for PMN 10TX13, Stephenville had similar results for the top producers.

Hemicellulose concentration was relatively uniform across all entries at each location; however, statistically significant differences occurred among the entries (Tables 6–8). Hemicellulose content of sunn hemp was significantly lower than all other entries (Tables 6 and 8). Cellulose concentrations were also mostly uniform in year two. The largest range occurred at Beeville, with BMR sorghum having the lowest concentration (341 g kg$^{-1}$) and sunn hemp having the highest concentration (477 g kg$^{-1}$). All other second year cellulose concentrations at all locations occurred within this range. BMR sorghum had the lowest ADL at all three locations, sunn hemp again produced the highest ADL. At Beeville, NDF was essentially equivalent for all entries except for BMR sorghum which was significantly lower than the others (Table 8). At Stephenville, BMR sorghum was also the lowest with Alamo switchgrass having the highest NDF (Table 6). At College Station, energy cane had the lowest NDF while the non-BMR annual sorghum SX-17 had the highest (Table 7). As in year one, sunn hemp had the highest crude protein percentage across all locations (Tables 6–8). Sunn hemp had the lowest silica concentrations as well as silica yields in year two. Also, as in year one for silica yield, biomass yield was more important than differences in silica content. The highest silica yields at each location were the same as the largest biomass yield at each location: PMN 10TX13 at College Station and Beeville, and Merkeron napiergrass at Stephenville (Tables 6–8).

**Table 6.** Second year Stephenville sample traits of biomass yield (kg ha$^{-1}$) and chemical composition including hemicellulose, cellulose, acid detergent lignin (ADL), neutral detergent fiber (NDF) (g kg$^{-1}$ biomass), crude protein (CP) (%), silica (%), and silica yield (kg ha$^{-1}$).

| Entry | Biomass | Hemicellulose | Cellulose | ADL | NDF | CP | Silica | Silica Yield |
|---|---|---|---|---|---|---|---|---|
| | kg ha$^{-1}$ | | g kg$^{-1}$ | | | % | | kg ha$^{-1}$ |
| (1) PMN 10TX13 [Y] | 17,226 cd [Z] | 270 abc | 392 abc | 66 abcd | 729 bcd | 2.9 bc | 3.73 a | 695 bcd |
| (2) Merkeron [X] | 60,950 a | 243 c | 412 abc | 77 abcd | 732 abcd | 2.96 bc | 2.48 a | 1486 a |
| (3) PEPU 09FL03 [W] | 55,315 a | 242 c | 366 c | 105 a | 713 cde | 2.08 cd | 2.89 a | 1338 ab |
| (4) PEPU 09FL01 [W] | 30,831 bc | 240 c | 437 a | 89 abc | 766 abcd | 1.45 d | 3.19 a | 867 abcd |
| (5) BMR Sorghum [V] | 8175 d | 269 abc | 359 c | 27 d | 654 e | 3.01 bc | 4.94 a | 405 d |
| (6) SX-17 [U] | 6006 d | 262 bc | 393 abc | 49 cd | 704 de | 1.86 cd | 3.6 a | 213 d |
| (7) BMR pearl millet [T] | 5144 d | 294 ab | 360 c | 63 abcd | 717 bcde | 3.62 b | 3.7 a | 183 d |
| (8) PSH 09TX15 [S] | 12,987 d | 291 ab | 432 ab | 63 abcd | 785 ab | 2.44 bcd | 5.05 a | 639 cd |
| (9) Alamo switchgrass | 15,905 cd | 307 a | 420 abc | 77 abcd | 804 a | 2 cd | 2.59 a | 442 d |
| (10) Tropical Isle Sunn Hemp | 10,636 cd | 145 d | 438 abc | 126 ab | 709 abcde | 7.71 a | | |
| (11) Giant miscanthus (Mxg) | 10,900 d | 294 ab | 414 abc | 71 abcd | 779 abc | 1.7 cd | 4.17 a | 461 cd |
| (12) Energy cane [R] | 44,015 ab | 278 abc | 368 bc | 51 bcd | 697 de | 2.6 bcd | 2.79 a | 1144 abc |

[Z] Means within a column under each main factor followed by the same letter (a,b,c,d) are not significantly different according to All Pairs Grouping, Tukey HSD. [Y] Pearl millet napiergrass hybrid PMN 10TX13. [X] Napiergrass accession. [W] Annual sorghum (BMR) cultivar. [V] Annual sorghum SX-17 cultivar. [U] Perennial sorghum hybrid PSH 09TX15. [T] Exceed BMR pearl millet. [S] Perennial sorghum hybrid PSH 09TX15. [R] Energy cane unknown accession.

**Table 7.** Second year College Station sample traits of biomass yield (kg ha$^{-1}$) and chemical composition including hemicellulose, cellulose, acid detergent lignin (ADL), neutral detergent fiber (NDF) (g kg$^{-1}$ biomass), crude protein (CP) (%), silica (%), and silica yield (kg ha$^{-1}$).

| Entry | Biomass | Hemicellulose | Cellulose | ADL | NDF | CP | Silica | Silica Yield |
|---|---|---|---|---|---|---|---|---|
| | kg ha$^{-1}$ | | g kg$^{-1}$ | | | % | | kg ha$^{-1}$ |
| (1) PMN 10TX13 [Y] | 62,912 a [Z] | 202 c | 424 abcd | 88 ab | 714 ab | 2.21 ab | 2.83 cd | 1893 a |
| (2) Merkeron [X] | 61,525 a | 224 bc | 430 abc | 97 a | 750 ab | 2.65 ab | 2.49 cd | 1552 ab |
| (3) PEPU 09FL03 [W] | 49,086 ab | 225 abc | 416 abcd | 97 a | 738 ab | 1.82 ab | 2.16 cd | 1048 abc |
| (4) PEPU 09FL01 [W] | 37,306 abc | 243 abc | 418 abcd | 101 a | 762 a | 1.97 ab | 2.52 cd | 917 abc |
| (5) BMR Sorghum [V] | 8238 c | 269 ab | 410 bcd | 51 c | 730 ab | 1.94 ab | 4.7 ab | 405 bc |
| (6) SX-17 [U] | 5330 c | 271 ab | 458 a | 78 abc | 808 a | 1.4 b | 3.09 bcd | 169 c |
| (7) BMR pearl millet [T] | 7465 c | 275 ab | 404 cd | 63 bc | 742 ab | 2.92 a | 3 bcd | 212 c |
| (8) PSH 09TX15 [S] | 6715 c | 273 ab | 449 ab | 71 abc | 792 a | 2.17 ab | 4.86 a | 334 bc |
| (9) Alamo switchgrass | 14,613 c | 287 a | 421 abcd | 83 abc | 791 a | 2.29 ab | 2.01 cd | 284 bc |
| (11) Giant miscanthus (Mxg) | 16,922 bc | 279 ab | 419 abcd | 69 abc | 767 a | 1.87 ab | 3.55 abc | 610 abc |
| (12) Energy cane [R] | 54,816 a | 230 abc | 380 d | 56 bc | 665 b | 1.99 ab | 1.4 d | 778 abc |

[Z] Means within a column under each main factor followed by the same letter (a,b,c,d) are not significantly different according to All Pairs Grouping, Tukey HSD. [Y] Pearl millet napiergrass hybrid PMN 10TX13. [X] Napiergrass accession. [W] Annual sorghum (BMR) cultivar. [V] Annual sorghum SX-17 cultivar. [U] Perennial sorghum hybrid PSH 09TX15. [T] Exceed BMR pearl millet. [S] Perennial sorghum hybrid PSH 09TX15. [R] Energy cane unknown accession.

**Table 8.** Second year Beeville sample traits of biomass yield (kg ha$^{-1}$) and chemical composition including hemicellulose, cellulose, acid detergent lignin (ADL), neutral detergent fiber (NDF) (g kg$^{-1}$ biomass), crude protein (CP) (%), silica (%), and silica yield (kg ha$^{-1}$).

| | | | | | | Means | | |
|---|---|---|---|---|---|---|---|---|
| **Entry** | **Biomass** | **Hemicellulose** | **Cellulose** | **ADL** | **NDF** | **CP** | **Silica** | **Silica Yield** |
| | **kg ha$^{-1}$** | | **g kg$^{-1}$** | | | **%** | | **kg ha$^{-1}$** |
| (1) PMN 10TX13 [Y] | 40,098 a [Z] | 261 bc | 392 bc | 51 bc | 704 a | 4.48 ab | 3.88 ab | 1562 a |
| (2) Merkeron [X] | 39,783 a | 257 bc | 395 bc | 63 b | 715 a | 3.42 b | 3.45 ab | 1410 ab |
| (3) PEPU 09FL03 [W] | 34,867 ab | 255 bc | 365 bc | 71 ab | 691 a | 4.26 ab | 4.15 ab | 1373 ab |
| (4) PEPU 09FL01 [W] | 30,298 abc | 247 c | 416 b | 82 ab | 745 a | 3.43 b | 2.67 bc | 855 abc |
| (5) BMR Sorghum [V] | 10,005 abc | 263 bc | 341 c | 22 c | 626 b | 4.35 ab | 6.08 a | 608 abc |
| (6) SX-17 [U] | 6345 bc | 278 abc | 375 bc | 56 bc | 708 a | 3.4 b | 5.54 a | 353 abc |
| (7) BMR pearl millet [T] | 3431 c | 295 abc | 394 bc | 56 bc | 745 a | 3.8 b | 4.93 ab | 177 bc |
| (8) PSH 09TX15 [S] | 5841 bc | 288 abc | 381 bc | 51 bc | 720 a | 4.78 ab | 5.63 a | 338 abc |
| (9) Alamo switchgrass | 11,543 abc | 299 ab | 366 bc | 56 bc | 721 a | 3.7 b | 3.7 ab | 430 abc |
| (10) Tropical Isle Sunn Hemp | 3401 c | 141 d | 477 a | 105 a | 723 a | 6.99 a | 0.49 c | 18 c |
| (11) Giant miscanthus (Mxg) | 7068 bc | 316 a | 350 c | 46 bc | 712 a | 4.2 ab | 6.13 a | 432 abc |
| (12) Energy cane [R] | 34,214 abc | 273 abc | 382 bc | 47 bc | 702 a | 3.41 b | 3.61 ab | 1229 abc |

[Z] Means within a column under each main factor followed by the same letter (a,b,c) are not significantly different according to All Pairs Grouping, Tukey HSD. [Y] Pearl millet napiergrass hybrid PMN 10TX13. [X] Napiergrass accession. [W] Annual sorghum (BMR) cultivar. [V] Annual sorghum SX-17 cultivar. [U] Perennial sorghum hybrid PSH 09TX15. [T] Exceed BMR pearl millet. [S] Perennial sorghum hybrid PSH 09TX15. [R] Energy cane unknown accession.

## 4. Discussion

Integrated biorefineries have diverse designs. One strategy utilizing biomass fractionation could produce ethanol from cellulose and hemicellulose, biopower from lignin, and silica from the remaining mineral fraction (Figure 2). Also, there could be the possibility of recovering other soluble products (crude protein) from the initial liquid fraction.

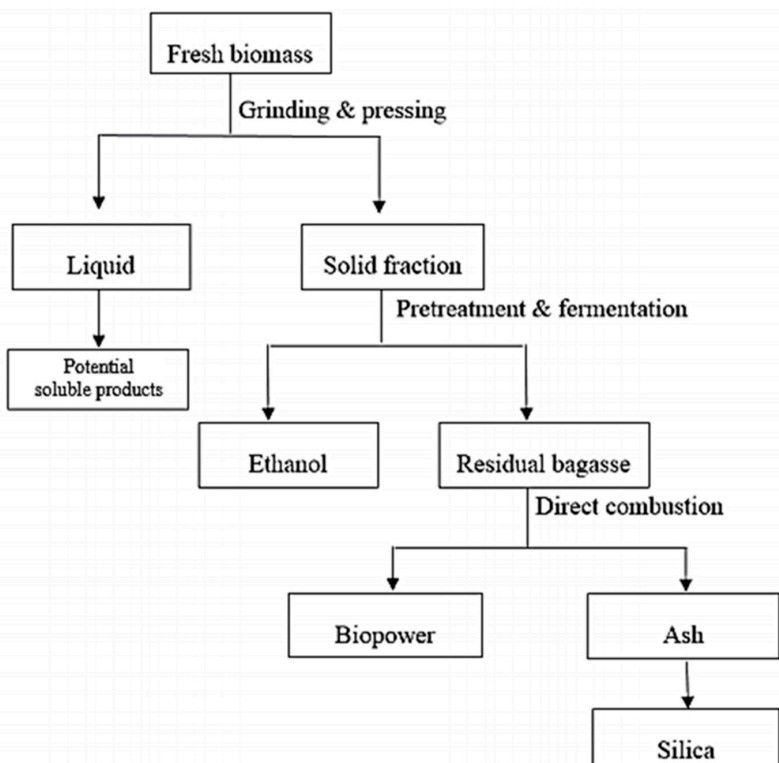

**Figure 2.** Illustration of conceptual integrated biorefinery producing ethanol, biopower, and silica.

Selection of feedstocks to maximize the yield of both biofuels and co-products is critical for the economic viability of integrated biorefineries. With cellulosic ethanol currently not economically competitive compared to fuel ethanol from sugar- and starch-based feedstocks [60–62], isolation of biosilica from the inorganic mineral fraction offers potential to provide additional revenue to biorefineries.

Biomass yield and forage quality was similar for several feedstocks in this study (napiergrass, PMN, and switchgrass) to previously published reports [20–22,26,34–36]. Our data for such across the other included feedstocks represent novel, baseline data for their performance across equivalent ecoregions. All of the Si content and yield data reported herein are novel across the feedstocks and adaptation zones. The two reports of XRF-based Si estimation in napiergrass [11,63] have ranged from somewhat lower to somewhat higher than those in this study; however, they were both performed in tropical regions and would not provide relevant yield estimates for subtropical and temperate regions. The single report of XRF-based Si content in switchgrass [64], was based only on bulk biomass samples of unknown origin and without the associated biomass yields.

In this study, summary rankings of the feedstocks were based on their overall suitability for utilization in the above proposed integrated biorefinery by four criteria: (1) increased overall biomass yield, (2) increased cellulose and hemicellulose contents, (3) reduced lignin content, and (4) increased silica content (Table 9). While recognizing that many considerations (costs for feedstock, pretreatment, saccharification, fermentation, ethanol recovery, etc.) are used when selecting a feedstock, our results show that the most promising feedstocks for an integrated biorefinery that captures value from both the lignocellulosic biomass and mineral fractions within ecoregions equivalent to the three in this study are PMN, napiergrass, and energy cane. Even though the biomass yields in this study for napiergrass, PMN, and switchgrass were similar to previous reports [20–22,26,34–36], additional larger scale trials would be warranted for all included feedstocks in all targeted production regions prior to commercialization. More extensive Si sampling of both raw biomass and post pre-treatment residues would also be beneficial.

**Table 9.** Summary ranking of potential biorefinery (bioethanol, biopower, biosilica) across twelve candidate feedstocks based on biomass (highest cellulose and hemicellulose, lowest ADL) and silica yields (kg ha$^{-1}$).

| Entry | First Year | | | Second Year | | |
|---|---|---|---|---|---|---|
| | College Station | Stephenville | Beeville | College Station | Stephenville | Beeville |
| (1) PMN 10TX13 | 1 | 1 | 1 | 1 | 5 | 1 |
| (2) Merkeron | 2 | 4 | 1 | 2 | 1 | 2 |
| (3) PEPU 09FL03 | 4 | 3 | 4 | 4 | 2 | 5 |
| (4) PEPU 09FL01 | 5 | 5 | 7 | 5 | 4 | 6 |
| (5) BMR sorghum | 6 | 10 | 8 | 7 | 9 | 4 |
| (6) SX-17 | 10 | 5 | 6 | 11 | 10 | 9 |
| (7) BMR pearl millet | 12 | 11 | 11 | 10 | 11 | 11 |
| (8) PSH 09TX15 | 9 | 7 | 5 | 9 | 6 | 9 |
| (9) Alamo switchgrass | 6 | 8 | 9 | 8 | 7 | 7 |
| (10) Tropical Isle Sunn Hemp | 6 | 8 | | | | 12 |
| (11) Giant miscanthus (Mxg) | 11 | 12 | 10 | 6 | 8 | 7 |
| (12) Energy cane | 3 | 2 | 3 | 3 | 2 | 3 |

PMN, napiergrass, and energy cane consistently had higher biomass yields across all locations. The yield results were not surprising, although as miscanthus typically takes two years to establish, it is expected that miscanthus could achieve higher yields in subsequent years. Whether miscanthus in later years would yield as high as the three grasses listed above is unknown. Sunn hemp had roughly equivalent biomass yields as the grass entries; however, due to its low silica yield, it would not be an

ideal feedstock for the proposed biorefinery. The highest biosilica yields were also recovered from PMN, napiergrass, and energy cane. Silica percentages were higher than previously reported mean or median concentrations, there is likely a substantial environmental effect at the test locations as silica concentrations trended higher across entries (except sunn hemp).

Energy cane and napiergrass must be propagated vegetatively and thus have higher establishment costs. Napiergrass is also considered an invasive species in some areas (Florida). PMN hybrids in contrast have lower establishment costs via direct seeding, as well as having no seed: weed invasiveness potential as a sterile triploid crop. These considerations, along with the biomass and silica data, indicate PMN is a strong candidate feedstock for integrated biorefineries. Additional evaluations are needed to determine its adaptability for other regions and environments. In this study, however, PMN hybrids consistently yielded greater than 1000 kg/ha of silica in College Station and Beeville (years 1 and 2) as well as in Stephenville in year 1. With market prices for amorphous silica at USD 800 per ton [65], PMN has significant potential towards integrated biofuel: silica biorefinery strategies and replacement of fume silica with renewable biosilica. Global silica markets currently have a 9% annual growth rate and will surpass USD 9 billion in the near future [66], further indicating economic opportunities for biosilica.

**Author Contributions:** Conceptualization, Y.X., B.L.B., and R.W.J.; methodology, Y.X., J.L.F., J.P.M., P.S., and R.W.J.; formal analysis, Y.X.; writing—original draft preparation, Y.X., N.P.; writing—review and editing, Y.X., N.P., J.L.F., J.P.M., P.S., B.L.B., and R.W.J.; supervision, R.W.J.; project administration, Y.X.; funding acquisition, R.W.J. All authors have read and agreed to the published version of the manuscript.

**Funding:** This research received no external funding.

**Conflicts of Interest:** The authors declare no conflicts of interest.

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
