# Peer review of "Silica Production across Candidate Lignocellulosic Biorefinery Feedstocks"

_agronomy, doi:10.3390/agronomy10010082_

Round 1
Reviewer 1 Report
In the submitted manuscript a very thorough analysis of the biomass composition of several biorefinery feedstock candidates is presented.
Recommendations for the minor corrections:
Line 53-54: bioproducts (bioplastics, etc.) from the remaining lignin…
Please provide a reference so the reader would be familiar with current situation in the field.
Line 67-68: Lignin can, however, be used to produce lignosulfonate.
Please rephrase. Lignin could not be used to produce lignosulfonates. Lignosulfonates are generated under a particular pulping conditions and lignin obtain from a different pulping procedures could not be transformed into lignosulfonates.
Line 97: Please, specify for a general reader what is “black liqour”
Author Response
Review 1 Recommendations
Line 53-54: bioproducts (bioplastics, etc.) from the remaining lignin…
Please provide a reference so the reader would be familiar with current situation in the field.
A reference has been added to the section that addresses the potential lignin based products.
Line 67-68: Lignin can, however, be used to produce lignosulfonate.
Please rephrase. Lignin could not be used to produce lignosulfonates. Lignosulfonates are generated under a particular pulping conditions and lignin obtain from a different pulping procedures could not be transformed into lignosulfonates.
Changed sentence to: If different pulp processing methods are used lignosulfonates can be produced in leu of lignin being directly combusted.
Line 97: Please, specify for a general reader what is “black liqour”
Added definition to sentence:
A two-step process to isolate lignin and silica from biomass derived black liquor – a waste product produced in the pulping process that is high in lignin and other extracts – revealed that high silica recovery in the precipitate could be achieved at pH 6-7.

Reviewer 2 Report
November 4, 2019
This manuscript addresses the potential biorefinery using twelve lignocellulosic biomass based on their chemical composition and silica yields. This is somehow interesting manuscript providing helpful information to the readers of Agronomy readers; however, the reviewer has the multiple queries listed below to improve the manuscript that should be replied in an exhaustive and sound basis before considering the article for publication.
There are some minor errors (font, font size, figures, tables, and formatting) in the whole manuscript that should be corrected. The authors present a general sugar-flat form process, silica, and twelve agricultural feedstock in introduction section, which is nicely summarized; however, these are too general and wordy to present the current work. The reviewer concerns that this part covers more than a half of the manuscript and dilute the original goal, purpose, and novelty of this research, the authors want to address. The reviewer understands why the authors analyzed the silica yield in each sample but is not sure how the authors estimate the production of biofuels, value-added chemicals, and potential utilizations. Even though the high number of chemical composition (mainly cellulose and hemicellulose) may provide more sugar yields, various pretreatment technique such as hydrothermal, chemical, dilute acid, ammonia, and others can result in actual sugar yields with a following enzymatic saccharification. The current study shows the difference of chemical composition and silica yield in each sample but it will not provide an increase of biorefinery yield. To provide better comparisons and difference in feedstock, the authors need to obtain actual sugar yields and/or biofuels with different biomass the authors have. There is no method how the authors measured the chemical composition of biomass samples. The main results of this work are biomass yield, chemical composition, and ranking of twelve feedstock for potential biorefinery. The reviewer wonders how the authors ranked twelve samples (just based on number of cellulose, hemicellulose, and silica composition). However, again, the reviewer believes that the increase of chemical composition would not reflect the increase of potential biorefinery. This will be clearly explained how and why the authors ranked twelve biomass samples. There are informative descriptions for comparing chemical composition of each sample but the authors missed the main connection between the results and novelty. The high number of chemical composition will be various in different years but it is not enough to address the novelty of the work.Author Response
This manuscript addresses the potential biorefinery using twelve lignocellulosic biomass based on their chemical composition and silica yields. This is somehow interesting manuscript providing helpful information to the readers of Agronomy readers; however, the reviewer has the multiple queries listed below to improve the manuscript that should be replied in an exhaustive and sound basis before considering the article for publication.
We greatly appreciate the guidance and have worked to address all of the noted concerns.
There are some minor errors (font, font size, figures, tables, and formatting) in the whole manuscript that should be corrected.
We have thoroughly reviewed the manuscript, identifying and correcting several additional minor grammatical and formatting errors.
The authors present a general sugar-flat form process, silica, and twelve agricultural feedstock in introduction section, which is nicely summarized; however, these are too general and wordy to present the current work. The reviewer concerns that this part covers more than a half of the manuscript and dilute the original goal, purpose, and novelty of this research, the authors want to address.
We have revised and shortened the Introduction section—in particular the feedstock descriptions (1.3.1 through 1.3.8)—by roughly 20%. This significant reduction should more closely approach the balance suggested by the reviewer between adequate background coverage and new findings in our research.
The reviewer understands why the authors analyzed the silica yield in each sample but is not sure how the authors estimate the production of biofuels, value-added chemicals, and potential utilizations. Even though the high number of chemical composition (mainly cellulose and hemicellulose) may provide more sugar yields, various pretreatment technique such as hydrothermal, chemical, dilute acid, ammonia, and others can result in actual sugar yields with a following enzymatic saccharification. The current study shows the difference of chemical composition and silica yield in each sample but it will not provide an increase of biorefinery yield. To provide better comparisons and difference in feedstock, the authors need to obtain actual sugar yields and/or biofuels with different biomass the authors have.
We agree that economic analyses of biofuel conversion platforms is difficult and largely unresolved to date. For this reason, our manuscript focuses instead on the potential for additional value and revenue in biorefineries through isolation and utilization of co-products such as biosilica from mineral fractions. This proposed strategy would not impact biomass fiber composition and therefore be neutral in its effect upon either conversion platform (bioethanol, biopower, syngas, etc.) or the mentioned pretreatments (physical, chemical, physio-chemical, biological). We understand that this point may not have been sufficiently emphasized, and we have added clarifying text to several sections (1.1, 1.2, 1.4, 4) as well as 3 relevant references (Clayton, 2019; Liu et al., 2019; Nuelle, 2019).
There is no method how the authors measured the chemical composition of biomass samples. The main results of this work are biomass yield, chemical composition, and ranking of twelve feedstock for potential biorefinery. The reviewer wonders how the authors ranked twelve samples (just based on number of cellulose, hemicellulose, and silica composition). However, again, the reviewer believes that the increase of chemical composition would not reflect the increase of potential biorefinery. This will be clearly explained how and why the authors ranked twelve biomass samples. There are informative descriptions for comparing chemical composition of each sample but the authors missed the main connection between the results and novelty. The high number of chemical composition will be various in different years but it is not enough to address the novelty of the work.
We have added clarifying text in section 4 regarding criteria for our ranking of the feedstocks. All chemical conversion methods used are described in section 2.2.3. As in our above response, the novelty and focus of our research was primarily on the potential value and revenue from biosilica capture from the typically unutilized mineral fraction in biofuel refineries. In this respect, our Si findings are not directly impacted by chemical composition. The composition and biomass yields were included instead as indicators of performance for readers—no economic analyses of bioethanol or other biofuels was intended.

Reviewer 3 Report
I believe the manuscript presents useful data, but I believe there are the manuscript should be modified prior to publication.
The assertion that silica can be economically produced from agricultural residues needs to be better supported. Is this really a viable process? In addition to reporting the yield results (both biomass & silica), these results need to be put in context. How do these yields compare to work the authors cite in section 1.3? Are there any surprising results?One other minor issue - clearly state how cellulose and hemicellulose were determined (cellulose = ADF - ADL, hemicellulose = NDF - ADF)
Author Response
Reviewer 3 recommendations
I believe the manuscript presents useful data, but I believe there are the manuscript should be modified prior to publication.
The assertion that silica can be economically produced from agricultural residues needs to be better supported. Is this really a viable process?
Silica is commonly being extracted from rice hull ash, it has yet to be done with bioenergy residue, but the process would be similar. We make no assertion on whether it is economically feasible for a bioenergy plant but only as a potential revenue source. Statements were added to clarify that no economic analysis was completed. We understand that this point may not have been sufficiently emphasized, and we have added clarifying text to several sections (1.1, 1.2, 1.4, 4) as well as 3 relevant references (Clayton, 2019; Liu et al., 2019; Nuelle, 2019).
In addition to reporting the yield results (both biomass & silica), these results need to be put in context. How do these yields compare to work the authors cite in section 1.3? Are there any surprising results?
We have added statements regarding biomass and silica yield connecting results to previous work in the discussion.
One other minor issue - clearly state how cellulose and hemicellulose were determined (cellulose = ADF - ADL, hemicellulose = NDF - ADF)
The formulas were added to 2.2.3 in materials and methods. The last sentence now reads:
Nitrogen (N) was determined using an elemental analyzer (Vario Macro, Elementar, Germany) and crude protein (CP), cellulose, and hemicellulose were calculated (CP = N, % DM basis x 6.25) (Cellulose = ADF – ADL) (Hemicellulose = NDF – ADF).

Round 2
Reviewer 2 Report
The revised version of manuscript addressed the reviewer’s comments and suggestions; however, the reviewer believes that this is not enough to be accepted to Agronomy journal. Please see further comments and suggestions below.
Check the font size, it should be 10 pt for the manuscript and 9 pt for tables and figures. Although the authors have modified and shortened the Introduction parts, it is too long and wordy that may dilute the whole manuscript concept. There are twelve samples the authors used here; however, only 8 crops are described in the manuscript. These may be grouped and will describe shortly with the key rationale why these types of crops are important. The reviewer agrees the concept on the potential for additional value in biorefineries using biosilica from mineral fractions. The reviewer concerns that the main calculations are carried out with a small amount of silica analysis, and following results are applied to the calculations to estimate the total silica concentration. In order to obtain accurate data and estimation, bench scale (at least) and pilot scale tests and estimations need to be performed and sequential results should be discussed for the estimations (kg/ha) There are some calculations and estimations with different year of biomass that provide interesting results and potential research; however, the results and discuss are not fully addressed with a high level description. There are some comparisons that which one is better or worse, but further potential reasons and comparisons with other similar works and/or studies with pretreated biomass to emphasize the novelty of the current work. The reviewer can image the workload and time the authors spent for this research; however, there are many published papers for biosilica productions and potential research with new concept, and the current form is not enough to present the different and novelty. The authors may choose some major biomass (3 – 5 crops), perform a scale-up process, and estimation/calculation/comparison with each other and/or other references will be better.Author Response
The revised version of manuscript addressed the reviewer’s comments and suggestions; however, the reviewer believes that this is not enough to be accepted to Agronomy journal. Please see further comments and suggestions below.
Check the font size, it should be 10 pt for the manuscript and 9 pt for tables and figures.
We have reviewed and formatted the correct font size throughout.
Although the authors have modified and shortened the Introduction parts, it is too long and wordy that may dilute the whole manuscript concept. There are twelve samples the authors used here; however, only 8 crops are described in the manuscript. These may be grouped and will describe shortly with the key rationale why these types of crops are important.
We appreciate the continued guidance and have combined all of the feedstocks into a single, integrated subsection versus 8 individual subsections. It has also been significantly shortened and is now approximately 1/3 the length (~40 lines) of the original submission (~110 lines).
The reviewer agrees the concept on the potential for additional value in biorefineries using biosilica from mineral fractions. The reviewer concerns that the main calculations are carried out with a small amount of silica analysis, and following results are applied to the calculations to estimate the total silica concentration. In order to obtain accurate data and estimation, bench scale (at least) and pilot scale tests and estimations need to be performed and sequential results should be discussed for the estimations (kg/ha)
We have added clarifying text to the Discussion section (~ lines 582-587) to help reinforce that our field trials were based on small plots—and that further, larger trials would be needed for yield verification at commercial scale in future efforts. Since the biomass yields obtained in our trials were similar to others for some of the same feedstocks (napiergrass, PMN, switchgrass, etc.) referenced, we believe our achieved biomass yields were sufficiently representative. Our Si samples were taken from bulk, composite samples from each entire plot and replicated as rigorously as in the original source article (Reidinger et al., 2012) and other subsequent citing articles. With our Si results for napiergrass being intermediate to those reported by Lima et al., 2014 (somewhat lower than ours) and Rezende et al., 2018 (somewhat higher than ours), we believe our methodology is sufficiently robust for supporting evidence for Si content in napiergrass, as well as initial reports for such in our other included feedstocks.
There are some calculations and estimations with different year of biomass that provide interesting results and potential research; however, the results and discuss are not fully addressed with a high level description. There are some comparisons that which one is better or worse, but further potential reasons and comparisons with other similar works and/or studies with pretreated biomass to emphasize the novelty of the current work.
We have added to the Discussion (~lines 558-570) and 2 references (Rezende et al., 2018; Morgan et al., 2015) to both compare to related, relevant publications and emphasize the novelty of our work. Our manuscript is the first report of XRF-based Si content and yield for any biomass feedstocks in U.S. (temperate) adaptation zones. Rezende et al., 2018 included napiergrass but was only in a tropical environment (Brazil). Lima et al., 2014 also included napiergrass but was again only in a tropical environment. Morgan et al., 2015 included switchgrass in the U.S., but it was from an anonymous feedstock source without any associated biomass yield data. We are not aware of any other XRF-based Si content and biomass yield studies specifically for biofuel feedstocks in temperate regions.
The reviewer can image the workload and time the authors spent for this research; however, there are many published papers for biosilica productions and potential research with new concept, and the current form is not enough to present the different and novelty.
While the Si XRF methodology source (Reidinger et al., 2012) has been cited more than 50 times, we found that only 3 (Lima et al., 2014; Morgan et al., 2015, Rezende et al., 2018) of these analyzed biomass/ biofuel crops. The remainder have been primarily on herbarium specimens, root/ soil samples, and horticultural crops. We have added text (and 2 references) as to how our manuscript is specifically differentiated from these 3 papers as in our response to the comment above. We have added additional text to this effect in the Rationale section as well.
The authors may choose some major biomass (3 – 5 crops), perform a scale-up process, and estimation/calculation/comparison with each other and/or other references will be better.
We agree that additional studies would be valuable but respectfully request consideration for publication in Agronomy based on our findings being completely novel for ‘XRF-based Si determinations in any biofuel feedstock in temperate regions’.
